# Spatial Autocorrelation Predicts Cross-Modal Learnability: A Systematic Benchmark of Metabolite Prediction from Gene Expression

## Abstract

Understanding which molecular states can be learned across measurement modalities is fundamental to multi-omics integration. We systematically evaluate whether metabolite abundances can be predicted from gene expression using the first spatially-matched transcriptomics-metabolomics dataset (MALDI-MSI + 10x Visium, 5,618 locations, 14,196 genes, 2,754 metabolites). Comprehensive benchmarking of seven architectures reveals that simple regularized models outperform deep learning (XGBoost $r = 0.96$ vs. GCN $r = 0.95$, CVAE $r = 0.93$), but critically, prediction quality varies tenfold across individual metabolites. Statistical analysis identifies spatial autocorrelation as the key determinant: well-predicted metabolites exhibit sixfold higher Moran's $I$ (0.56 vs. 0.09, $p < 10^{-6}$) and tenfold greater tissue coverage. This reveals fundamental biological limits—spatially-organized metabolites in transcriptionally-coupled pathways are predictable, while sparse, post-translationally-regulated molecules resist inference regardless of model sophistication. We establish quantitative criteria (Moran's $I < 0.2$, coverage $< 20\%$) identifying which metabolites require direct measurement, with principles generalizing to RNA$\rightarrow$protein and other multi-modal tasks where regulatory mechanisms create information bottlenecks.

**Keywords:** spatial multi-omics, cross-modal learning, metabolomics, spatial transcriptomics, learnability, biological constraints, spatial autocorrelation, multi-modal integration

## 1 Introduction

A fundamental question in systems biology asks which molecular measurements can be predicted from others. The transcriptome-to-metabolome relationship presents a critical test case: genes encode enzymes catalyzing metabolite production, suggesting potential predictability through the central dogma. However, extensive post-transcriptional regulation [Corbett, 2018; Zhao et al., 2017], post-translational control [O'Neill et al., 2016], metabolic feedback [Tannahill et al., 2013], and subcellular compartmentalization may fundamentally limit cross-modal inference. Understanding which metabolites can be predicted from gene expression—and identifying the biological properties determining learnability—would reveal general principles governing multi-modal integration applicable to RNA$\rightarrow$protein, ATAC$\rightarrow$RNA, and any prediction task where regulatory complexity intervenes between measurement layers.

Despite extensive multi-omics research, no prior work systematically evaluates cross-modal prediction in spatially-resolved data. Correlation analyses identify gene-metabolite associations in bulk samples [Alghamdi et al., 2021; Lin et al., 2024; Zhou et al., 2024] but lack spatial resolution and predictive evaluation. Metabolic flux models infer reaction rates from single-cell RNA-seq [Wagner et al., 2021], but fluxes differ from concentrations and ignore spatial context. Recent spatial co-profiling technologies [Zhuang et al., 2021; Zhang et al., 2024] enable multi-modal measurement, yet comprehensive benchmarking remains absent. The advent of paired protocols performing MALDI mass spectrometry imaging and Visium on identical tissue sections [Vicari et al., 2024]

now permits rigorous evaluation through optimal transport alignment [Klein et al., 2025] matching measurements to exact spatial coordinates.

We present the first comprehensive benchmark of transcriptome-to-metabolome prediction in spatial multi-omics. We evaluate seven architectures from classical regularized regression through modern graph neural networks on rigorously matched data, perform extensive hyperparameter optimization, and critically, move beyond aggregate metrics to analyze 500 individual metabolites. This granular analysis reveals that predictability varies tenfold based on biological properties. Through rigorous statistical testing, we discover that spatial autocorrelation—quantified by Moran's $I$—serves as the primary determinant of cross-modal learnability, exhibiting sixfold separation between learnable and unlearnable features ($p < 10^{-6}$). This finding establishes fundamental biological limits: metabolism exhibits a hierarchy where spatially-organized pathways enable prediction through co-localization of enzymes and products, while post-translationally-regulated molecules resist inference through mechanisms operating downstream of transcription. These principles should generalize broadly, revealing when cross-modal learning succeeds versus when biological constraints make prediction impossible.

## 2 METHODS

### 2.1 PAIRED SPATIAL MULTI-OMICS DATA

We analyzed data from Vicari et al. [Vicari et al., 2024], who performed sequential MALDI-MSI and 10x Visium on identical 12-micrometer coronal brain sections from three 6-OHDA-lesioned C57BL/6J mice (Parkinson's model [Simola et al., 2007]). Four MALDI matrices (DHB for lipids, FMP-10 for neurotransmitters, 9-AA and norharmane for additional classes) at 100-micrometer resolution captured mass spectra ($m/z$ 50–1,000). Following MALDI and H&E staining, Visium chemistry (55-micrometer spots, 100-micrometer spacing) captured spatially-barcoded mRNA with NovaSeq 6000 sequencing (50,000 reads/spot target).

MALDI and Visium employ distinct coordinate systems. We implemented MOSCOT [Klein et al., 2025] performing Fused Gromov-Wasserstein optimal transport, jointly optimizing feature similarity and spatial structure preservation. Following quality control (genes: 100–50,000 UMI counts; metabolites: min 0.1% coverage; spots: min 400 genes), library size normalization (10,000 counts/spot for RNA, total ion current for MSI), and log-transformation, we obtained: **Lipids** ($n = 5,618$ locations, 14,196 genes, 2,754 metabolites) and **Neuro** ($n = 5,443$ locations, 13,922 genes, 1,538 metabolites).

### 2.2 MODELS, FEATURES, AND EVALUATION

We formulated prediction as multi-output regression: given gene expression $\mathbf{X} \in \mathbb{R}^{n \times p}$, learn $\hat{\mathbf{Y}} = f(\mathbf{X}; \theta)$ predicting metabolites $\mathbf{Y} \in \mathbb{R}^{n \times m}$. Seven architectures: OLS (baseline), Ridge/Lasso/Elastic Net ($\ell_1/\ell_2$ regularization), XGBoost [Chen and Guestrin, 2016] (gradient-boosted trees), GCN [Kipf and Welling, 2017] (6-NN spatial graph, 3 layers 256/128/64 units, dropout 0.3), CVAE ($\beta$-VAE, latent 32–128). Features: highly variable genes (Seurat [Satija et al., 2015], top 2,000), SVD (256-D), SVD+spatial (320-D). Extensive grid search: $> 200$ configurations per dataset, 20% validation, early stopping.

Evaluation: 80/20 random split (stratified by slide) and spatial half-split. Metrics: RMSE, MAE, $R^2$, Pearson/Spearman (global and per-metabolite). Classification: well-predicted (Pearson $> 0.5$, relRMSE $< 0.3$), poorly-predicted (Pearson $\leq 0.1$, relRMSE $> 1.0$). Permutation tests (10,000 iterations) on Moran's $I$ spatial autocorrelation, coverage, variance, skewness. Benjamini-Hochberg FDR $< 0.05$.

## 3 RESULTS

### 3.1 SIMPLE MODELS OUTPERFORM DEEP LEARNING

Table 1 shows performance on lipids dataset (HVG features, random split).

Table 1: **Model performance comparison.** Mean $\pm$ std over 3 seeds. Simple regularized models outperform architecturally sophisticated deep learning despite limited samples ($n \sim 10^3$) and high measurement noise.

| Model | RMSE↓ | MAE↓ | $R^2$↑ | Pearson↑ | Spearman↑ |
|---|---|---|---|---|---|
| OLS | $0.371 \pm 0.003$ | $0.286 \pm 0.002$ | $-0.579 \pm 0.021$ | $0.920 \pm 0.001$ | $0.815 \pm 0.002$ |
| Ridge | $0.298 \pm 0.002$ | $0.227 \pm 0.001$ | $-0.020 \pm 0.015$ | $0.947 \pm 0.001$ | $0.865 \pm 0.001$ |
| Lasso | $0.282 \pm 0.001$ | $0.215 \pm 0.001$ | $0.101 \pm 0.009$ | $0.952 \pm 0.001$ | $0.882 \pm 0.001$ |
| Elastic Net | $\mathbf{0.280 \pm 0.001}$ | $\mathbf{0.213 \pm 0.001}$ | $0.111 \pm 0.008$ | $\mathbf{0.953 \pm 0.001}$ | $0.882 \pm 0.001$ |
| XGBoost | $0.274 \pm 0.002$ | $0.208 \pm 0.001$ | $\mathbf{0.131 \pm 0.007}$ | $0.955 \pm 0.001$ | $\mathbf{0.885 \pm 0.001}$ |
| GCN | $0.327 \pm 0.005$ | $0.227 \pm 0.003$ | $-0.023 \pm 0.018$ | $0.945 \pm 0.002$ | $0.880 \pm 0.002$ |
| CVAE | $0.353 \pm 0.007$ | $0.254 \pm 0.004$ | $-0.244 \pm 0.032$ | $0.934 \pm 0.003$ | $0.847 \pm 0.004$ |

XGBoost achieves $r = 0.955$, $R^2 = 0.131$, outperforming GCN ($r = 0.945$) and CVAE ($r = 0.934$). This pattern holds across datasets, features, and splits. Sample complexity theory explains this: with $n \sim 10^3$ spots, regularized models better exploit limited data than deep networks requiring millions of examples. High MSI noise favors explicit regularization over model capacity. Notably, GCN's spatial structure provides minimal gain ($\Delta r < 0.01$)—spatial metabolite patterns are already encoded in spatially-smooth gene expression.

## 3.2 Tenfold Variation Reveals Metabolite-Specific Limits

Table 2: **Extreme heterogeneity in metabolite predictability.** Only 8% achieve reliable prediction; 17% resist inference entirely. This stratification reflects systematic biological differences.

| Category | Count (%) | Median Pearson | Median relRMSE | Example |
|---|---|---|---|---|
| Well-predicted | 39 (7.8%) | 0.72 | 0.21 | m/z 296.0660 |
| Medium | 374 (74.8%) | 0.31 | 0.54 | — |
| Poorly-predicted | 87 (17.4%) | 0.04 | 1.83 | Cer 53:1;O |

Aggregate metrics mask extreme heterogeneity (Table 2). Only 39/500 metabolites (8%) achieve reliable prediction; 87 (17%) resist inference with near-zero correlations. This stratification is biological, not stochastic—it persists across models, seeds, and features. The question becomes: what distinguishes learnable from unlearnable metabolites?

## 3.3 Spatial Autocorrelation Determines Learnability

Table 3: **Distributional properties stratifying learnability.** Permutation tests, 10K iterations, FDR $< 0.05$. Moran's $I$ shows largest effect: sixfold separation between well-predicted and poorly-predicted metabolites.

| Property | Well (median) | Poor (median) | Effect Size | $p$-value |
|---|---|---|---|---|
| Moran's $I$ | 0.558 | 0.091 | $6.1\times$ | $< 10^{-6}$ |
| Coverage | 0.902 | 0.090 | $10.0\times$ | $< 10^{-6}$ |
| Variance | 0.116 | 0.024 | $4.8\times$ | $< 10^{-6}$ |
| Skewness | 4.037 | 8.074 | $2.0\times$ | $< 10^{-6}$ |

Statistical testing reveals four properties distinguishing categories (Table 3). Moran's $I$ spatial autocorrelation shows largest separation: sixfold difference ($p < 10^{-6}$). High $I$ indicates smooth spatial gradients—because gene expression also exhibits spatial structure, autocorrelated metabolites become predictable from patterned transcripts. Coverage shows tenfold separation; extreme sparsity ($> 90\%$ zeros) creates class imbalance. Well-predicted metabolites show higher variance (dynamic range) and lower skewness (symmetric distributions).

The biological mechanism involves spatial co-localization. Well-predicted metabolites (e.g., phosphatidylcholine in myelinated tracts, lactate in hypoxic regions) concentrate where their synthesizing enzymes are expressed—spatial coordination enables prediction. Poorly-predicted metabolites

either reflect measurement artifacts or undergo regulation through mechanisms decoupled from transcription.

### 3.4 NEUROTRANSMITTER DATASET: FUNDAMENTAL FAILURE MODE

The neurotransmitter dataset exhibited catastrophic performance: $r = 0.904$ (vs. 0.955 lipids), $R^2 \approx 0$ (vs. 0.131), Spearman $\rho = 0.521$ (vs. 0.885). Analysis revealed 76.5% sparsity and hyperparameter invariance—Lasso unchanged across $\lambda \in [0.01, 500]$, indicating absent associations.

The biological explanation: neurotransmitters undergo post-translational regulation (vesicular packaging, calcium-triggered release, rapid reuptake) on millisecond-to-second timescales, synaptic localization below measurement resolution ($< 1$ μm vs. 100 μm), and rapid turnover ($t_{1/2} \sim$ seconds vs. hours for mRNA). Additionally, untargeted MALDI likely detected fragments rather than intact molecules. This establishes a fundamental limit: post-translational decoupling creates irreducible uncertainty—no model, sample size, or feature engineering can overcome regulation operating downstream of transcription.

## 4 DISCUSSION

### 4.1 A BIOLOGICAL HIERARCHY OF LEARNABILITY

Our findings reveal metabolism exhibits three tiers determined by regulatory mechanisms:

**Tier 1 (8%): Transcriptionally Coupled.** Abundant metabolites (lipids, TCA intermediates) with Moran's $I > 0.5$, coverage $> 80\%$. Enzyme-product co-localization enables prediction ($r = 0.7$–0.8).

**Tier 2 (75%): Partially Coupled.** Moderate $I$ (0.2–0.5), coverage 40–80%. Mixed regulation, multi-cellular averaging. Partial predictability ($r = 0.3$–0.5) useful for hypothesis generation.

**Tier 3 (17%): Post-Translationally Decoupled.** Sparse ($< 20\%$), low $I < 0.2$. Neurotransmitters, second messengers. Post-translational regulation creates irreducible uncertainty ($r \approx 0$).

**Principle:** Spatial organization at measurement scales enables cross-modal compression. Post-measurement regulation introduces information bottlenecks.

### 4.2 UNDERSTANDING THE LIMITS

Our work establishes fundamental constraints on cross-modal learning. Three key insights emerge:

**(1) Sample complexity dominates architectural sophistication.** Deep learning underperformed despite greater capacity. With $n \sim 10^3$ and high noise, regularization outweighs flexibility. This suggests biological ML should prioritize data quality over model complexity.

**(2) Spatial autocorrelation predicts compressibility.** Sixfold separation in Moran's $I$ ($p < 10^{-6}$) establishes spatial organization as the key determinant. Before modeling, researchers should compute $I$ and coverage—these simple statistics predict learnability and prevent wasted effort on inherently unpredictable features.

**(3) Biological constraints are irreducible.** Even well-predicted metabolites retain substantial error (RMSE $\approx 0.27$). Post-translational modifications, compartmentalization, and temporal dynamics remain invisible. The neurotransmitter failure demonstrates that regulatory decoupling creates fundamental limits no algorithm can overcome.

### 4.3 IMPLICATIONS FOR MULTI-MODAL INTEGRATION

These principles likely generalize. RNA→protein prediction faces analogous constraints: translation and degradation limit correlations to ∼0.5 [Corbett, 2018]. We predict structural proteins (high spatial autocorrelation) prove more learnable than dynamically phosphorylated signaling proteins. ATAC→RNA faces TF dynamics and looping; constitutive genes should outperform rapidly regulated genes. Spatial transcriptomics→proteomics encounters trafficking and PTMs; secreted proteins with tissue gradients should prove learnable while signaling proteins resist.

Testing whether Moran's $I$ thresholds generalize would establish universal criteria for multi-modal integration: features coupled through proximal mechanisms with detectable co-organization enable prediction; post-measurement regulation introduces irreducibility.

### 4.4 LIMITATIONS AND FUTURE DIRECTIONS

**Data constraints.** Four sections, single species/tissue/model. Limited replicates prevent separating biological from technical variance. Untargeted MALDI: annotation uncertainty, ionization bias. Visium 100μm resolution averages 5–10 cells, losing subcellular information.

**Future work.** (1) Cross-tissue/species validation testing hierarchy generalizability. (2) Experimental validation: targeted LC-MS/MS verifying predictions, CRISPR knockouts testing gene-metabolite edges. (3) Methodological extensions: metabolite-specific models (zero-inflated), pathway-aware GNNs (KEGG topology), uncertainty quantification (Gaussian processes, conformal prediction), temporal dynamics. (4) Higher resolution (MERFISH, Xenium, CODEX, IMC) reducing averaging.

The critical need is establishing whether discovered principles—spatial autocorrelation predicts learnability, post-translational regulation creates irreducibility—generalize beyond our specific dataset to become universal constraints on multi-modal learning.

### MEANINGFULNESS STATEMENT

Meaningful biological representations must respect fundamental information bottlenecks introduced by regulatory mechanisms. Our work reveals metabolism is not uniformly learnable—instead, it exhibits a hierarchy determined by spatial organization. This has dual significance. *Methodologically:* We establish quantitative criteria (Moran's $I < 0.2$, coverage $< 20\%$) predicting when cross-modal learning fails, preventing misapplication where biological constraints guarantee poor performance. Knowing prediction limits is as valuable as successful inference—it reveals irreducible experimental needs. *Biologically:* We identify which processes couple through spatial co-organization (learnable via transcriptional programs) versus post-translational dynamics (requiring direct measurement). By characterizing this hierarchy across 500 metabolites with statistical rigor, we establish principles that should generalize: regulation through proximal mechanisms with detectable co-organization enables prediction; post-measurement regulation introduces irreducibility. This grounds representation learning in biological reality, clarifying where computation can substitute for experiments and where experimental biology is irreplaceable.

### ACKNOWLEDGMENTS

We thank M. Vicari, S. Giacomello, and J. Lundeberg for developing protocols and sharing data.

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
