# OpenReview forum: "Spatial Autocorrelation Predicts Cross-Modal Learnability: A Systematic Benchmark of Metabolite Prediction from Gene Expression"
_ICLR.cc/2026/Workshop/LMRL — ICLR 2026 Workshop LMRL Poster_

### Official Review · Reviewer_qzjD · 2026-02-22
**Interesting yet limited benchmark on metabolite-gene expression cross-modal learning**

**Rating:** 5
**Confidence:** 3

**Review:**

This paper presents a systematic benchmark of predicting spatial metabolite abundances from spatial gene expression using paired MALDI-MSI and 10x Visium data from Vicari et al. The authors evaluate seven models (regularized regression, XGBoost, GCN, CVAE) and show that simple regularized methods outperform deep learning ones. Crucially, prediction quality varies dramatically across metabolites, and the authors identify spatial autocorrelation as the primary determinant of cross-modal learnability. The paper proposes quantitative thresholds (e.g., Moran’s I < 0.2) to flag inherently unlearnable targets.

Strengths:
- Clear and systematic benchmarking across multiple architectures.
- Empirical finding that simple regularized models outperform more complex deep learning approaches under realistic sample sizes.
- Interesting identification of spatial autocorrelation as a predictor of cross-modal learnability.

Weaknesses:
- The benchmarked models are largely standard multi-output regression approaches. While this supports the authors’ claim that biological structure dominates architectural sophistication, the modeling suite does not include more structured or pathway-aware models that might better exploit gene-metabolite relationships. Thus, the conclusion that learnability limits are “fundamental” may be somewhat overstated.
- Spatial smoothness may inflate performance under random splits; stronger spatial generalization tests would strengthen the claims.
- The work is based on a single dataset (one tissue, one model system), limiting generalizability of the proposed Moran’s I thresholds.
- For a full 8-page submission, the empirical depth feels limited: results are summarized in only three tables with no figures, qualitative visualizations, or detailed analyses. This makes the paper feel somewhat shallow relative to a full-paper expectation.

Overall: A conceptually strong and biologically meaningful study that identifies spatial autocorrelation as a key determinant of cross-modal learnability. However, the modeling breadth and empirical depth are somewhat limited for a full-paper submission, and broader validation would be needed to support claims of general biological principles. The work would be better suited for a tiny paper - preliminary work submission.

---

### Official Review · Reviewer_tf9A · 2026-02-24
**Promising Dataset, but Underdeveloped Benchmark**

**Rating:** 3
**Confidence:** 3

**Review:**

**Minor Comments:**

- MALDI should be defined on first use, with a brief explanation of what MALDI-MSI measures and why it's relevant to this task
- The Methods section reads as a compressed list of decisions rather than a coherent narrative. Consider restructuring it to walk the reader through the pipeline sequentially: data acquisition --> alignment --> preprocessing --> model formulation --> evaluation. Each step should explain not just what was done listed sporadically, but why. Consider including supporting figures or schematics in future iterations of this paper as well. It'll help the reader get a better understanding of what is exactly being done.
-  What split is Table 1 corresponding to? Should be explicitly stated in the caption.

**Major Comments:**

1. **Dataset description is insufficient for the paper's core contribution.** The dataset is positioned as the primary contribution, yet its description is scattered and surface-level. The Methods should include a dedicated, self-contained subsection covering: tissue source and experimental conditions, the sequential MALDI-MSI and Visium protocol (what each modality captures, at what resolution, and how they differ), the alignment procedure via MOSCOT (and what it is/does), and the final data dimensions after QC. MOSCOT in particular is mentioned without explanation—readers need to understand what it does and why this is necessary here. A figure illustrating the spatial layout of matched spots would substantially aid comprehension.

2. **The prediction task itself is never clearly stated.** It is not obvious from the paper what the input and output of each model actually are. Are models trained per-metabolite or jointly? What does a single data point look like? This should be made explicit early in the Methods, ideally with a schematic or concrete example (e.g., "given a 14,196-dimensional gene expression vector at a spatial location, predict a 2,754-dimensional metabolite abundance vector"). You mention things briefly but not with enough detail in the paper.

3. **The train/test splits are insufficient for a rigorous benchmark.** Random and spatial half-splits alone do not adequately probe generalization in a dataset with this level of multimodal structure. Given that the dataset spans transcriptomic, metabolic, and spatial variation simultaneously, the benchmark would benefit from additional evaluation splits including: gene-based splits (e.g., held-out GO term categories to test whether models generalize across functional gene groups), metabolite-based splits (e.g., using Tanimoto similarity or scaffold-based clustering to ensure structurally distinct metabolites are held out), and pathway-specific splits (e.g., holding out entire metabolic pathways to assess whether pathway-level biology is being captured or merely interpolated). Without these, it is difficult to determine whether strong performance reflects genuine cross-modal learning or artifacts of spatial or molecular proximity between train and test examples.

4. **The model selection rationale is absent.** The paper benchmarks seven architectures but does not justify why these were chosen or what design space they are meant to cover. Are these representative of the current state of the field? Why were no pretrained or foundation model-style approaches included, given recent interest in large-scale omics models? A brief discussion of the design principles behind the model selection (and acknowledgment of what is excluded and why) would strengthen the benchmarking framing.

---

### Official Review · Reviewer_9c2Q · 2026-02-25
**High-value benchmark and negative-result paper on cross-modal learnability, with strong evidence but overstated “fundamental limits” claims**

**Rating:** 8
**Confidence:** 4

**Review:**

Summary: The paper benchmarks prediction of metabolite abundance from spatial gene expression using a paired spatial transcriptomics + metabolomics setup (e.g., Visium + MALDI-MSI aligned data), asking not only which model performs best, but which metabolites are learnable and why.

What it proposes/finds:

1. Across seven model families (linear/regularized models through deeper architectures), simple regularized models perform competitively or better than deep models in this data regime.

2. Aggregate performance hides substantial metabolite-level heterogeneity: only a subset are well predicted, while many are poorly predicted.

3. Spatial autocorrelation (Moran’s I) and coverage are strong predictors/associates of learnability, with large separations between well- and poorly-predicted metabolites.

4. The neurotransmitter case serves as a compelling biologically grounded negative result / failure mode.

Main claims:

1. Cross-modal learnability is metabolite-specific and strongly heterogeneous.

2. Spatial autocorrelation is a dominant predictor (or correlate) of learnability in this benchmark.

3. Biological/regulatory constraints can limit what models can learn across modalities.

Therefore, benchmarking should move beyond aggregate metrics and incorporate feature-level learnability diagnostics. Overall, this is a strong benchmark/diagnostic workshop paper with a valuable negative result and a genuinely useful biological insight (Moran’s I stratification), but broader “fundamental limits” and cross-modality universality claims should be toned down.

Quality (Technical Soundness): This is a strong empirical paper overall, especially in breadth of comparison and feature-level statistical analysis.

What is shown:

1. A clear paired spatial multi-omics prediction setup with documented preprocessing/alignment.

2. Benchmarking of multiple model classes with multiple metrics and multiple split strategies (including spatially structured splits, which is important in spatial data).

3. Per-metabolite stratification over a large subset (e.g., 500 metabolites), showing substantial heterogeneity in predictability.

4. Strong statistical evidence that spatial autocorrelation (Moran’s I), and to a degree coverage, separates well- vs poorly-predicted metabolites (including permutation-based significance testing / multiple-testing control, as described).

5. A well-characterized neurotransmitter failure mode that is informative rather than dismissed.

What is inferred (reasonable but not fully proven):

1. The biological interpretation that spatially organized / transcriptionally coupled metabolites are more learnable is plausible and well motivated by the Moran’s I results.

2. The discussion of post-translational regulation and spatial-scale mismatch as causes of failure (especially for neurotransmitters) is biologically plausible and useful for hypothesis generation.

What is overstated / needs calibration:

1. “Fundamental biological limits,” “irreducible uncertainty,” and especially formulations like “no model/sample size/feature engineering can overcome this” are too strong for the current evidence.

2. The study is observational and dataset-specific.

3. It does not systematically vary sample size, assay resolution, or alternative measurement platforms.

4. Generalization claims to other modalities (e.g., RNA→protein, ATAC→RNA) are plausible but untested in this submission and should be framed as hypotheses/implications rather than established findings.

Uncertainty / robustness / leakage-risk comments:

Strengths: The paper includes seed variability reporting and statistical testing for the key stratification claims. Also, spatially structured split evaluation is important and should reduce optimism from spatial leakage relative to random splits.

Weaknesses: The main narrative should more prominently compare random vs spatial split outcomes, especially given that spatial autocorrelation is central to the story. Additionally, the per-feature thresholding used to define “well-/poorly-predicted” groups appears reasonable, but sensitivity analysis for those thresholds should be shown more explicitly.

Metric interpretation caveat:

If the paper reports very high Pearson correlations alongside weak/negative R^2  in some settings, this deserves explicit explanation in the main text to avoid reader confusion (the metrics answer different questions and can diverge substantially under calibration/variance mismatch). Overall, the evidence for the core benchmark insight is strong; the main issue is claim calibration, not technical execution.

Clarity:

1. The paper is generally well written and well organized.

2. The move from aggregate metrics to per-metabolite learnability analysis is especially well motivated and clearly communicated.

3. The framing of a learnability hierarchy is intuitive and well thought out.

Areas to improve: Make random-vs-spatial split robustness more prominent in the main results narrative. More clearly separate: statistical associations (Moran’s I ↔ learnability), biological hypotheses (enzyme-product co-localization, post-translational decoupling), and stronger impossibility/fundamental-limit claims. The authors should clarify metric interpretation (especially Pearson vs R^2, if both are heavily used), in future work.

Originality: The paper’s novelty lies in a benchmark + diagnostic reframing, not a new predictor architecture.

Important original contributions include: feature/metabolite-level learnability analysis rather than aggregate benchmark-only reporting, identifying Moran’s I as a practical predictor/correlate of cross-modal learnability, and delivering a strong, biologically interpretable negative result (neurotransmitter failure). This is a strong LMRL-style contribution because it asks when biological signals are representationally learnable across modalities—not just which model scores highest.

Significance
1. Practical significance: High. The paper could change how multimodal prediction studies are designed and evaluated: evaluate per-feature learnability, report spatially robust splits, and pre-screen targets using properties like Moran’s I / coverage before committing modeling effort. The simple-models-vs-deep-models result is also a valuable corrective in this regime.

2. Scientific significance: Moderate-to-high. The paper provides a meaningful empirical link between learnability and measurable biological/spatial structure. However, it is not yet evidence for universal limits across tissues/species/modalities.

LMRL relevance: Very strong. The paper directly addresses when learned cross-modal representations can be meaningful and when biological structure imposes limits on representational transferability.

Pros:

1. Strong benchmark paper with a genuinely informative negative result.

2. Excellent feature-level (metabolite-level) analysis instead of relying on aggregate metrics.

3. Moran’s I learnability stratification is novel, practical, and biologically interpretable.

4. Good breadth of model comparisons and evaluation settings.

5. Includes spatially structured split evaluation (important for leakage concerns).

6. Neurotransmitter failure analysis is a valuable and well-discussed biological failure case.

Cons:

1. “Fundamental limits / irreducible uncertainty / impossible to overcome” language is over-strong for the current evidence base.

2. Generalization to other modalities (RNA→protein, ATAC→RNA, etc.) is speculative in this submission.

3. Main text should foreground random-vs-spatial split comparisons more clearly.

4. Threshold-based well/poor metabolite grouping needs explicit sensitivity analysis.

5. Biological mechanism discussion is plausible but mostly inferential (not directly validated).

6. Limited scope of biological systems (single tissue/species/platform context) constrains external validity.

Recommendation Rationale: The paper’s strongest contribution—a feature-level learnability benchmark showing that spatial autocorrelation strongly stratifies transcriptome-to-metabolome predictability, alongside an informative biological negative result—is directly shown and well supported. I am not rating it even higher because several key narrative claims are overstated relative to the current evidence and because robustness details (especially spatial split emphasis and threshold sensitivity) should be more central. With workshop calibration and scope, this is overall a strong submission for LMRL.

---

### Meta-Review · Area_Chair_muBV · 2026-02-25

**Recommendation:** Accept (Poster)
**Confidence:** 4

**Metareview:**

I recommend acceptance

---

### Decision · Program_Chairs · 2026-03-02

**Decision:**

Accept (Poster)

**Comment:**

Please see the meta-review.